# Raman Handheld Versus Microscopic Spectroscopy for Estimating the Post-Mortem Interval of Human Bones: A Comparative Pilot Study

**DOI:** 10.3390/bioengineering11111151

**Published:** 2024-11-15

**Authors:** Johannes Dominikus Pallua, Christina Louis, Nicole Gattermair, Andrea Brunner, Bettina Zelger, Michael Schirmer, Jovan Badzoka, Christoph Kappacher, Christian Wolfgang Huck, Jürgen Popp, Walter Rabl, Claudia Wöss

**Affiliations:** 1Department of Orthopaedics and Traumatology, Medical University of Innsbruck, Anichstraße 35, 6020 Innsbruck, Austria; johannes.pallua@i-med.ac.at (J.D.P.); christina.louis@student.i-med.ac.at (C.L.); niki@gattermair.at (N.G.); 2Institute of Pathology, Neuropathology, and Molecular Pathology, Medical University of Innsbruck, Muellerstrasse 44, 6020 Innsbruck, Austria; andrea.brunner-veber@innpath.at (A.B.); bettina.zelger@i-med.ac.at (B.Z.); 3Office Dr. Schirmer, 6060 Hall, Austria; schirmer.michael@icloud.com; 4Institute of Analytical Chemistry and Radiochemistry, University of Innsbruck, Innrain 80-82, 6020 Innsbruck, Austria; jbadzoka@gmail.com (J.B.); christoph.kappacher@uibk.ac.at (C.K.); christian.w.huck@uibk.ac.at (C.W.H.); 5Leibniz Institute of Photonic Technology, 9, 07745 Jena, Germany; juergen.popp@leibniz-ipht.de; 6Institute of Forensic Medicine, Medical University of Innsbruck, Muellerstraße 44, 6020 Innsbruck, Austria; walter.rabl@i-med.ac.at

**Keywords:** human skeletal remains, post-mortem interval, Raman, principal component analyses

## Abstract

The post-mortem interval estimation for human skeletal remains is critical in forensic medicine. This study used Raman spectroscopy, specifically comparing a handheld device to a Raman microscope for PMI estimations. Analyzing 99 autopsy bone samples and 5 archeological samples, the research categorized them into five PMI classes using conventional methods. Key parameters—like ν_1_PO_4_^3−^ intensity and crystallinity—were measured and analyzed. A principal component analysis effectively distinguished between PMI classes, indicating high classification accuracy for both devices. While both methods proved reliable, the fluorescence interference presented challenges in accurately determining the age of archeological samples. Ultimately, the study highlighted how Raman spectroscopy could enhance PMI estimation accuracy, especially in non-specialized labs, suggesting the potential for improved device optimization in the field.

## 1. Introduction

The determination of the post-mortem interval (PMI) based on bones is an essential subject for forensic medicine and archeology. The PMI must be determined quickly to establish whether human skeletal remains are of archeological interest or forensic relevance. More recent skeletal remains in question, which are less than a few decades old, are critical for prosecution, with the exact time frame varying according to the legal system of each country [1,2,3].

The determination of the PMI remains a substantial challenge [4]. Hitherto, existing methods have been either too inaccurate, destructive, or lacking in objectivity [5].

Taphonomy, i.e., the study of the state of an organism from the time of death to the time of discovery, comprises two fundamental sub-processes: decomposition and diagenesis. In the decomposition process, body and soft tissues are broken down by biological and microbiological activities until the organism is skeletonized. The duration of this process is highly dependent on environmental conditions and internal factors like health conditions, dietary habits, age, and others. The onset of skeletonization marks the beginning of diagenesis. Physical and chemical factors, such as pH value, humidity, temperature, soil pressure, oxygen content, and UV light in the post-mortem environment, lead to the demineralization of bones. Other taphonomic processes likely to occur are exposure to insects, other invertebrates, and scavenging animals [6,7,8].

It is also essential to differentiate between the spongiosa and the cortical bone to understand the taphonomical processes in bones. Spongiosa contains the organic phase, whose main components are collagen, lipids and proteins, while the cortical bone layer contains minerals responsible for the bone’s hardness, such as magnesium and hydroxyapatite [9,10,11]. During diagenesis, structural changes occur in these two bone components. The causes of the altered mineral structure and the increasing porosity of the bone include the breakdown of collagen, mineral and hydroxyapatite dissolution, ion exchange, and recrystallization. This susceptibility to diagenetic changes is different for the bone components [4,12,13,14]. Specific knowledge of the taphonomic processes and their influence on skeletal remains is essential to better understand bone degradation, especially during early PMI, i.e., when of forensic relevance [13].

The main reason for the relatively low number of experimental studies and research is that many methods for PMI determination are complex, expensive and, at the same time, unreliable. So far, most forensic research studies applied two established approaches to study PMI. One approach is based on real case studies in which a known taphonomic variable influences the PMI. Most of these studies applied methods limited to the phase of soft tissue decomposition, i.e., the early stages of taphonomy and its influencing factors [15,16]. Another approach comprises so-called body farms, where human or animal bodies are placed in a specific environment or under particular circumstances, helping to obtain a better understanding of the effects of specific taphonomic variables on body remains. These experiments also focus on soft tissue degradation and are conducted over less than one year [15,17,18]. Therefore, separate taphonomic data from archeological and palaeontological studies are used to assess PMIs longer than centuries or a thousand years [19,20,21,22,23]. With these studies in mind, it became clear that generalizing and creating realistic models of taphonomic processes in both soft tissue and bones should be critically perceived.

Most studies employed chemical or isotopic methods, which must be validated for PMIs of less than ten years. Additionally, the effect of weathering and changes in bone colour have also been employed in PMI estimation, with the influence of scavengers also considered [24,25,26,27]. These data were then extrapolated for shorter PMIs, resulting in an ongoing research gap for more accurate estimates of PMIs [2,3,28,29,30,31]. It would be beneficial for future studies to focus on models that are specifically designed to examine the effects of certain environmental factors on different tissues [32].

Raman spectroscopy (RS) has recently emerged as a valuable method for analyzing variations in the chemical composition of bone based on specific burial conditions. Studies indicate a consistent decline in the intensity of organic Raman bands, which are specific spectral features associated with organic compounds correlated with increased burial time [33]. Furthermore, RS has been effectively utilized to estimate PMI by exploring the relationship between bone mass and PMI [34]. In a controlled experiment, human ribs, devoid of soft tissue, were subjected to various environmental conditions over 90 days. An analysis of the Raman spectra, which demonstrated its adaptability in different ecological conditions, revealed the normalization of peaks associated with organic constituents within the bone throughout this three-month duration. Notable alterations included the broadening and attenuation of the amide I band, alongside changes in the CH_2_ region and the NH band.

Additionally, RS has shown its utility in assessing the PMI of skeletal remains by investigating the comprehensive chemical transformations occurring in bones and teeth, thereby establishing a link to the burial timeframe [33,35,36]. RS offers the advantages of being microscopic, rapid, and non-destructive (when the sample is small enough to fit the equipment), which are crucial for forensic or anthropological analysis, especially considering the availability of portable devices [11,33,37,38]. The RS technique provides a range of parameters for assessing bone tissue, including crystallinity, mineral-to-mineral ratios, mineral-to-matrix (MMR) ratios, mineral carbonate content (MinCarb), overall mineral content (MinCont), carbonate-to-phosphate ratios, and collagen cross-link ratios [39]. These Raman parameters yield critical insights into bone quality by examining its compositional and nanostructural characteristics. In this context, “quality” encompasses the tissue’s compositional integrity and architectural features, essential for resistance to deformation and fracture [40]. Pathological modifications, such as bacterial infection or mechanical compromise, can alter these bone quality parameters. RS can detect these alterations, thus providing valuable information about bone health and integrity [40].

Consequently, RS is an essential tool for analyzing the mineral and organic components of normal and altered bone tissue, with discussions of critical parameters—such as crystallinity, mineral-to-matrix ratio, and carbonate-to-phosphate ratio—detailed in [41]. This study has two primary objectives: first, to validate existing remote sensing (RS) data for PMI estimation, and second, to assess the effectiveness of a handheld Raman spectrometer versus a Raman microscope for PMI determination in human skeletal remains. Utilizing principal component analysis (PCA) to interpret the spectral data, this approach offers a non-destructive, refined methodology that advances traditional PMI estimation techniques. By clearly defining the originality of our approach, we expand on previous work and present specific solutions to current challenges in PMI research.

## 2. Materials and Methods

### 2.1. Human Bone Samples

This study systematically collected 99 contemporary forensic bone samples for molecular genetic identification during autopsies at the University Institute of Forensic Medicine. Five medieval archeological bone samples were also sourced from five European excavation sites. The post-mortem interval (PMI) classification class intervals were calibrated by §57 of the Austrian criminal code [42]. PMI categorization was determined based on collaborative law enforcement investigations and forensic requirements before applying near-infrared (NIR) spectrometry. The samples were distributed into the following PMI classes: class 1 (0–2 weeks, *n* = 32), class 2 (2 weeks–6 months, *n* = 46), class 3 (6 months–1 year, *n* = 11), class 4 (1–10 years, *n* = 10), and class 5 (>100 years, *n* = 5). These samples encompassed 16 female and 88 male remains. In ambiguous conventional PMI estimates, the average value was adopted for classification. We utilized standard anthropological methods [43,44]—such as analyses of pelvic and cranial features—and DNA typing for sex estimation. For analytical procedures, the diaphysis of the femur from both forensic and archeological samples was utilized. A 7 mm thick transverse section was longitudinally excised from each femur using a hand saw, removing the periosteum and bone marrow. The samples were air-dried for several days at ambient temperature. Reflectance spectroscopy (RS) was conducted before any subsequent forensic analyses. The study adhered to ICH-GCP guidelines (ICH Official web site: ICH) and the Declaration of Helsinki, with ethical approval secured from the local ethics commission (EK: 1357/2021).

### 2.2. Handheld Measurements

Raman spectra were acquired using the state-of-the-art Mira Raman handheld system (Metrohm Inula GmbH, Wien, Austria). Orbital raster scanning was employed in a circular pattern, with a spot size of 42 µm, covering a measurement area of 0.332 mm^2^. The polarization degree was set at 1000:1, and the system operated at a wavelength of 785 nm. The spectral range spanned from 2300 to 400 cm^−1^, with a spectral resolution of 6 cm^−1^. Each sample underwent analysis with five spectra collected from distinct locations. Measurements were performed at a controlled temperature of 22 °C and regulated humidity levels.

### 2.3. Micro-Raman Spectrtoscopy

Raman microscopic measurements were conducted in reflectance mode at ambient temperature using a Senterra II microscope (Bruker, Ettlingen, Germany). Excitation was achieved with a 785 nm nominal wavelength laser, operating at a power of 25 mW as measured at the back aperture of a Zeiss EC EPIPLAN 20x/0.4 objective (Carl Zeiss GmbH, Jena, Germany). Spectral analysis was performed over a range of 3600 cm^−1^ to 200 cm^−1^, with a resolution of approximately 4 cm^−1^. Before each measurement, the integration time for individual scan points was optimized near the scanning area to ensure a favourable signal-to-noise ratio while preventing sample damage. Randomized regions of interest (ROIs) were selected on the bone samples, and for each ROI, 36 spectra were captured, showcasing a high spatial resolution of 5 µm × 5 µm.

### 2.4. Data Processing

Data processing and image reconstruction were carried out using the OPUS 8.5 software (Bruker, Ettlingen, Germany). Data analysis was conducted using Unscrambler X 10.5 (AspenTech, Bedford, MA, USA), applying a reduction factor of 36, a 15-point Savitzky–Golay smoothing algorithm, and area normalization techniques. Area normalization was essential for comparative analyses of peak areas across samples, such as I958 or amide-I, particularly in heterogeneous porous materials like bone, where the excitation volume may vary, and beam focus can be inconsistent [45]. The peak intensity (I) [46,47,48,49,50,51] was evaluated through a specialized Microsoft Excel spreadsheet designed for analyzing both spectroscopic and chromatographic data [52], enabling the assessment of diagnostics. Once the preferred peak was chosen in the “Output” section, the baseline was adjusted through a linear subtraction using the x1–y1 and xn–yn coordinates. This approach facilitates the extraction of the maximum height (H) and the peak area, enhancing the versatility of data analysis. Two distinct area calculations are employed: the first, outlined in Equation (1), involves the partial summation of peak areas (*A*), while the second, described in Equation (2), computes the total intensity sum (Asi) [52].
(1)$$A=ai+aj+\ldots +an,\;being\;an=\frac{\left(yn-1+yn\right)\times (xn-1-xn)}{2}$$
(2)A=y1+y2+…+yn

The spectral parameters were analyzed statistically using GraphPad Prism software (version 9, San Diego, CA, USA). Comparisons were made employing the two-sample *t*-test, with significance determined at a *p*-value threshold of less than 0.05.

### 2.5. Principal Component Analyses: Modelling of Spectral Data

Principal Component Analyses (PCA) models were developed using Unscrambler X 10.5 by importing spectral data and implementing specific data pretreatments. The pretreatment protocol involved a reduction factor of 36, the application of 15-point Savitzky–Golay smoothing, and area normalization to optimize the spectral input for analysis.

## 3. Results

The aim of this study was to perform a comparative assessment of the efficacy of handheld Raman spectroscopy versus Raman microscopy in estimating post-mortem intervals (PMI) in human skeletal remains. A total of 104 samples were analyzed, comprising 16 samples from females and 88 from males. The PMIs were stratified into five categorical classes: 0 to 2 weeks (class 1, *n* = 32), 2 weeks to 6 months (class 2, *n* = 46), 6 months to 1 year (class 3, *n* = 11), 1 year to 10 years (class 4, *n* = 10), and exceeding 100 years (class 5, *n* = 5). Figure 1 has been included in the study to depict the advantages and disadvantages of traditional PMI estimation, Raman handheld, and Raman microscopy. It is evident from the research that the conventional PMI estimation method can be time-consuming, resource-intensive, and laborious in comparison to Raman.

### 3.1. Spectroscopy Data Evaluation

Bone tissue consists of two fundamental components: the organic matrix, predominantly composed of collagen fibres and non-collagenous proteins, and the inorganic phase, primarily comprising hydroxyapatite crystals, which provide rigidity and strength to the skeletal structure [53]. RS is a powerful tool for studying bones. It can measure things like the highest points or the size of areas under peaks at certain wavelengths [54]. The Raman bands detected in bone spectra correspond to phosphates, carbonate bone minerals, organics like collagen–proline, the hydroxyproline matrix, phenylalanine, amide I, II, and III, and CH-aliphatics [41]. Fluorescence interference is prevalent in Raman spectroscopy, particularly when assessing biological specimens like bone tissue. This phenomenon can complicate the interpretation of spectral data, as endogenous fluorophores in the samples may obscure the Raman signal, thereby affecting the accuracy of the analysis [55]. For instance, samples in class 5 PMI have a high level of single fluorescence, making them unsuitable for analysis with either the Raman handheld device or the Raman microscope (see Figure 2).

Figure 3 shows the mean value spectra of classes 1 to 4 and reveals PMI class-specific profiles. Prominent spectral features of classes 1 to 4 are phosphate (PO_4_^3−^), carbonate bone mineral (CO_3_^2−^), amide III, CH_2_ deformations (vibration) of protein and amide I of collagen. The phosphate (_ν4_PO_4_^3−^) at 577 cm^−1^ and the C-H groups ∼2800 cm^−1^ to 3000 cm^−1^ could only be detected with the Raman microscope. The significant differences between the 4 PMI classes are _ν2_PO_4_^3−^ at 450 cm^−1^, ν_1_PO_4_^3−^ at 958 cm^−1^, CO_3_^2−^ at 1070 cm^−1^, amide III at 1246 cm^−1^, CH_2_ at 1450 cm^−1^ and amide I at 1656 cm^−1^. The analysis reveals a marked deterioration in bone quality, specifically within the spectral range of 427 cm^−1^ to 577 cm^−1^. Additionally, alterations in protein conformation are evident, particularly at the peaks of 1001 cm^−1^ and in the region of 2800 to 3000 cm^−1^, though these changes are only observable using a Raman microscope.

In order to obtain accurate diagnostic information regarding changes in the course of the PMI, Raman spectra parameters are calculated for Raman handheld and Raman microscope. The present information describes the results of the derived spectral markers of both investigated devices for comparing the characterization of the investigated human bones in PMI classes 1 to 4. The markers include the intensity of the band (I), the area of the band (A) and the total width at half maximum (FWHM). Diagnostic significance was assessed using a simple one-way ANOVA (see Figure 4 and Figure 5 and Table 1 and Table 2) with a significance level of *p* < 0.05.

#### 3.1.1. Raman Spectra Parameters for Raman Handheld

The results show that ν_1_PO_4_^3−^ intensity has a significant difference between PMI class 1 and II (*p* = 0.0479 *), while there is no significant difference between PMI class 1 and 3 (*p* = 0.09553 ns) or 4 (*p* = 9731 ns). The crystallinity (1/FWHM) also shows a significant difference between PMI class 1 and 2 (*p* = 0.0154 *), but no significant difference between PMI class 1 and 3 (*p* = 0.5954 ns) or 4 (*p* = 0.5195 ns). The mineral/matrix ν_1_PO_4_^3−^/amide I ratio shows a significant difference between PMI class 1 and 2 (*p* < 0.0013), as well as between PMI class 1 and 3 (*p* = 0.0458), but no significant difference between PMI class 1 and 4 (*p* = 0.6382 ns). The mineral quality and crystallinity of the CO_3_^2−^/ν_1_PO_4_^3−^ ratio shows a significant difference between PMI class 1 and 2 (*p* = 0.0008), as well as between PMI class 1 and 4 (*p* < 0.0001 ****), but no significant difference between PMI class 1 and 3 (*p* = 0.3056 ns). The mineral carbonate content (MinCarb) shows no significant difference between PMI class 1 and 2 (*p* = 0.1292 ns), 3 (*p* = 0.0686 ns), or 4 (*p* = 0.9960 ns). The mineral-to-matrix ratio (MMR) and the CH-aliphatic content (CHACont) were not assessable. For amide I, the *p*-values based on a one-factor ANOVA indicate extremely high statistical significance for class 1 vs. 2 (*p* < 0.0001 ****), very high significance for class 1 vs. 3 (*p* = 0.0008 ***), and high significance for class 1 vs. 4 (*p* = 0.0026 **).

#### 3.1.2. Raman Spectra Parameters for Raman Microscope

The analyses show that there is a significant difference in the mineral/matrix ratio of ν_1_PO_4_^3−^/amide I when comparing PMI classes 1 and 4, although the significance level is very low (*p* = 0.0001 **** for PMI class 1 vs. 2. *p* = 0.0010 *** for PMI class 1 vs. 4). In addition, there are also significant differences in mineral quality and crystallinity of the CO_3_^2−^/ν_1_PO_4_^3−^ ratio and MinCarb between PMI classes 1 and 4 (*p* < 0.05). The results also show a significant difference in amide I between PMI classes 1 and 2, and 3 and 4 (*p* < 0.01). For the amount of ν_1_PO_4_^3−^, the *p*-values based on a one-factor ANOVA indicate no significant difference for class 1 vs. 2 (*p* = 0.2948) and class 1 vs. 3 (*p* = 0.7299), but a significant difference for class 1 vs. 4 (*p* = 0.0347 *). From the mineral component amount to the organic one (A958/A1656), the *p*-values indicate extremely high significance for class 1 vs. 2 (*p* < 0.0001 ****), very high significance for class 1 vs. 3 (*p* = 0.0003 ***), and very high significance for class 1 vs. 4 (*p* = 0.0010 ***). In summary, Raman microscope-derived spectral markers can help characterize human bone by distinguishing between different PMI classes. In particular, the mineral/matrix ratios of ν_1_PO_4_^3−^/amide I, the mineral quality and crystallinity of carbonate/phosphate and the mineral carbonate content appear to be useful markers to distinguish PMI classes. However, it should be noted that some markers were not assessable and further studies are required to confirm and extend the results.

The findings indicate that employing handheld Raman devices instead of Raman microscopes for bone analysis is a promising approach and may serve as a diagnostic tool for PMI characterization. However, it should be noted that not all spectral markers show significant differences between PMI classes and that further specific studies are required to confirm the accuracy and reliability of this method.

### 3.2. Diagnostic Performance PCA

Our experimental analysis involved employing PCA to study the average spectra of bone samples corresponding to different PMI categories, ranging from one to four, to detect changes in bone tissue. However, we excluded class five from our analysis due to poor data quality due to fluorescence. The possible diagnostic value of PCA applied to spectroscopic data has been shown in earlier research [36,56,57,58,59,60,61,62,63]. We employed PCA to evaluate the averaged spectra obtained from the Raman handheld device and the Raman microscope. Our investigation encompassed 104 bone samples, adhering to the methodological approaches established in prior research [36,61,64]. Implementing PCA in diagnostic methodologies is essential for achieving high accuracy and efficiency in bone tissue characterization. Our approach, which is accurate and efficient, involved generating a loadings plot to identify principal components, effectively showcasing the primary sources of variance within the dataset. This visualization enables a detailed examination of the correlations among individual variables, facilitating a deeper understanding of the underlying structure of the data [65]. The analysis is represented through correlation coefficients, from a maximum of 1 to a minimum of −1, illustrated graphically in Figure 6. The zero line delineates points of equivalence across the PMI classes. Variations in the positive and negative ranges effectively facilitate the distinction among the three principal components. This experimental study confirms and reiterates that such differentiation is feasible, providing reassurance about the study’s conclusions. We used a loading plot to express the relationship between the original and the new variables. The most significant variance within the dataset was due to ν_1_PO4^3−^, CO_3_^2−^, amide III, CH_2_ deformation, amide I, and C-H stretching. These regions can be used to identify spectral regions that allow for the better separation of the samples.

Table 3 comprehensively analyses eight specific wavenumber ranges employed in PCA.

The PCA models indicate that both methodologies’ most informative spectral data are predominantly found in regions I, IV, and V, as illustrated in Figure 7. The score plots reveal a remarkably strong correlation between principal components PC1 and PC2 across PMI classes 1 to 4 within these regions, alongside the corresponding loadings for PC1. Notably, PC1 explains the variance of 91%, 84%, and 93% for the Raman handheld spectrometer and 94%, 62%, and 85% for the Raman microscope across the spectral regions of PO4^3−^ (427 cm^−1^), amide III, and amide I, as represented in Figure 7 for regions I, IV, and V. Both methodologies demonstrate reliability and robustness, with significant potential for further refinement in estimating post-mortem intervals (PMIs). The automated and objective nature of PCA renders it a valuable tool for the routine assessment of PMIs in bone samples. This approach could substantially enhance PMI estimation capabilities for laboratories lacking specialized expertise.

## 4. Discussion

Regarding the estimation of PMI, the present study demonstrates that the quality of the data obtained from the Raman handheld device and the Raman microscope is comparable for non-archeological bones (as evidenced by Table 1 and Table 2 for bones less than 100 years old). This comparison is based on the intensity, area, and total width of band signals in the Raman spectrum caused by different organic and inorganic bone constituents. Therefore, RS enables the comprehensive examination of the spectral characteristics associated with pivotal key molecular structures, including phosphates, carbonates, collagen, and assorted protein-related markers, with ν_1_PO_4_^3−^, crystallinity, MMR, mineral quality and crystallinity, MinCarb, amide I, and CHACont, which reflects the diverse degradation processes of the bones over time.

In detail, Raman microscopy results showed significant differences in MMR as an essential indicator for bone quality between PMI classes 1 and 3 and 3 and 4, while the CI showed no significant differences between PMI groups. These results are consistent with the current literature [36,60]. Both systems identify the ν_1_PO_4_^3−^ phosphate group as an essential discriminating factor between the PMI classes and amide I and III as important spectral features. Concerning the ν_1_PO_4_^3−^ content of bone tissue, whose band is located at 960 cm^−1^, as a marker for the mineral content in the bone matrix, both Raman methods showed differences only between PMI classes 1 and 2, and not between PMI classes 1 and 3 or 4, which contradicts a prior observation that the ν_1_PO_4_^3−^ content decreases with increasing post-mortem time [54]. Further analyses included the 1/FWHM describing the crystallinity of the bone mineral and the amide I band reflecting the protein content in the bone matrix, which both decreased with increasing PMI. Others described that the local environment influences MinCont and MinCarb in bone for about 2 years, and the MinCarb of buried bone decreases steadily between 3 and 23 months [66]. In this study, the MinCarb remained relatively constant between 4 and 24 months, which can be explained by a different local environment. We can only speculate that the various temperatures and humidity of the surroundings resulted in the decreasing amount of MinCarb in the Howes et al. samples, as they were all buried [66].

The spectral quality of the Raman spectra from both instruments was similar. However, the Raman handheld instrument showed a higher signal-to-noise ratio and a lower spectral background than the Raman microscope. The Raman handheld device has a limited measuring range of 2300 to 400 cm^−1^, while the Raman microscope measures 3600 cm^−1^ to 200 cm^−1^. Thus, the Raman microscope has a broader coverage of the Raman spectrum and provides more spectral features for PCA. The PCA analyses show that both the Raman handheld instrument and the Raman microscope are suitable for identifying and quantifying spectral differences between the PMI classes, although both systems extract different components to differentiate the PMI classes. Taken together, both systems identify similar spectral features as important discriminating factors and could, therefore, be used as promising techniques to characterize PMI of human bone. The only difference is that as the Raman microscope’s PCA results show a higher variance resolution for the PC1 components than the Raman handheld device, the Raman microscope provides higher accuracy in PMI characterization. Also, Raman microscopy offers higher spatial resolution and can produce more optimized Raman images of the bone samples.

Regarding spectral parameters, the Raman handheld showed higher ν_1_PO4^3−^, amide I, and MinCarb intensities but lower MMR CHAcont and 1/FWHM values than the Raman microscopy. The Raman microscope also identified several additional spectral features, such as CH strains and CO_3_^2−^, which the Raman handheld device did not detect. These minor differences in the spectral parameters between the two instruments can most likely be attributed to the different instrument-specific factors, such as laser power, collection optics, etc.

When comparing the two instruments in terms of ease of use, the Raman handheld device offers a faster and more convenient way to analyze bone samples, as it is portable and allows for real-time analysis. Both RS measurement methods are considered non-invasive, meaning the samples’ integrity can be preserved. Conventional methods for determining the PMI are usually less objective, more expensive, and time-consuming, so RS is a well-suited technique for PMI estimation [67]. Thus, the overall results of this work show that RS is a fast, non-destructive, sensitive, and promising method for characterizing human bones, especially in determining PMI as a diagnostic marker for forensic medicine. Analytic techniques such as the PCA method support the interpretation of large amounts of data generated by RS, which is needed to establish the applicability of this method in forensic medicine.

The most important limitation of the RS technique was that the archeological bone samples of PMI class 5 did not provide usable spectra and could not be included in the analysis to compare the Raman instruments—Raman handheld and Raman microscopy. Fluorescence presents a significant challenge in Raman spectroscopy, as it overlays and obscures the Raman signal, particularly in PMI class 5 samples—those older than 100 years. These samples exhibit elevated fluorescence levels, challenging practical analysis with handheld or microscope-based Raman devices. Fluorescence in bone arises primarily from the organic and inorganic components within the bone matrix, such as collagen and various proteins, which tend to fluoresce under light exposure. Over time, as bones degrade, organic by-products increase, leading to higher fluorescence in older samples [55]. In contrast, more recent samples contain fewer fluorescent compounds, making fluorescence interference less critical for non-archeological bones. While several techniques, including UV filters, specific laser wavelengths, and fluorescence suppression algorithms, can mitigate fluorescence [55], no such methods were applied in this study. Future research could enhance the analysis of older bone samples by integrating fluorescence suppression techniques, which would expand the applicability of Raman spectroscopy for a broader range of forensic samples.

Further limitations of the study are that despite the relatively large sample size, there is still uncertainty regarding the exact PMI, as PMI classification can only be based on police and forensic investigations. This PMI is sometimes imprecise, as, for example, a bone sample with an estimated age of 14 days (+/−1 day) can be assigned to both PMI class 1 and PMI class 2. Therefore, if the conventional estimate was uncertain, the average PMI result was used. Additional reasons for the incorrect categorization of the PMI classes caused by the underlying lack of a precise PMI include the individual chemical composition of the bones, unequal locations, thermal effects, environmental influences, and different decomposition processes. Another limiting factor is the pre-mortem condition of the bones, as an altered bone structure in diseases such as osteoporosis or other bone-destructive metabolic or infectious diseases falsifies the results. Pathological changes can affect the mineral and organic content, altering MMR, amide, and crystallinity readings. Future studies should account for these variables by including samples with known pathological conditions and assessing how they influence RS data to ensure more accurate PMI classification. The results of RS measurements may vary significantly depending on environmental conditions, particularly when samples are exposed to non-controlled outdoor settings with diverse soil compositions, pH, moisture, and temperature variations. Simulating these conditions in future studies could provide more comprehensive data on how environmental factors affect RS measurements. Additionally, comparative analysis using other optical methods, such as Fourier-transform infrared spectroscopy, could further validate RS findings and improve reliability across various environments. PCA was chosen for its ability to reduce dimensionality and identify significant patterns within spectral data. However, we acknowledge that other machine learning techniques, such as support vector machines (SVMs) or random forest classifiers, could improve classification efficiency. Future studies could incorporate these methods to evaluate their effectiveness in enhancing PMI classification accuracy compared to PCA.

Despite all these imprecise comparisons, the PCA of Raman data was able to differentiate between the PMI classes 1 to 4. This success underscores the potential of Raman spectroscopy as a valuable addition to the forensic toolkit, complementing other methods such as DNA analysis, toxicology, and traditional post-mortem interval estimation techniques. Raman spectroscopy offers unique advantages for non-destructive sample analysis, particularly in cases where sample preservation is critical. By identifying organic and inorganic markers within bone tissue, Raman spectroscopy could be used with techniques like radiocarbon dating or isotopic analysis to verify or enhance PMI assessments. The portability of the handheld Raman spectroscopy device further enhances its value for in-field forensic investigations, potentially allowing for initial on-site analysis before more detailed examinations in the lab. When combined with statistical methods like PCA, Raman spectroscopy supports the efficient processing of large datasets and generates reliable and reproducible PMI estimates. This multidisciplinary integration represents a significant advancement in forensic science, promising faster and more precise analysis in forensic investigations.

## 5. Conclusions

This study demonstrates the effectiveness of Raman spectroscopy in estimating the post-mortem interval (PMI) for human skeletal remains. While both the handheld and Raman devices provide valuable spectral data, the microscope’s broader spectral range and higher spatial resolution enhance PMI characterization by capturing finer details within bone samples. However, the handheld device’s portability and rapid data acquisition make it a promising tool for preliminary in-field forensic assessments. Both instruments successfully identify key spectral markers, such as phosphate, carbonate, and amide bands, allowing for differentiation across PMI classes in non-archeological samples. The potential of Raman spectroscopy to complement traditional forensic methods lies in its ability to non-destructively analyze organic and inorganic bone markers, offering a fast, reliable, and non-invasive approach to PMI assessment. Future research should address fluorescence challenges in older bones, incorporate machine-learning techniques, and evaluate the influence of environmental factors on RS measurements. This study supports using Raman spectroscopy as a valuable addition to the forensic toolkit, especially when integrated with PCA, to streamline data interpretation and enhance PMI estimation accuracy.

## Figures and Tables

**Figure 1 bioengineering-11-01151-f001:**
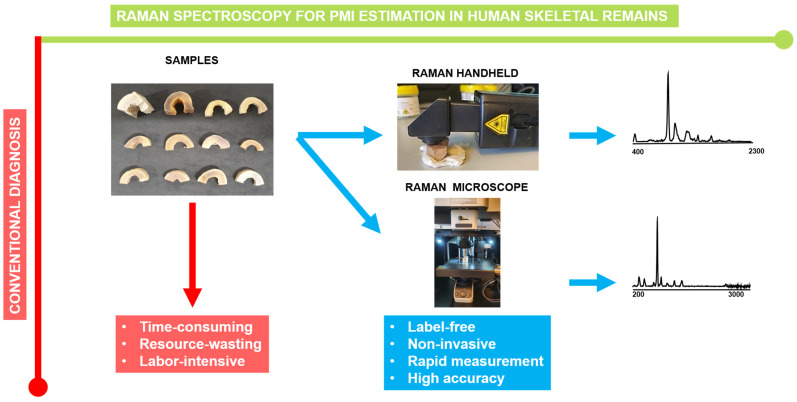
A comparison of RS and conventional PMI estimation, highlighting their pros and cons. Spectra from the Raman handheld and Raman microscope were depicted in the experiment.

**Figure 2 bioengineering-11-01151-f002:**
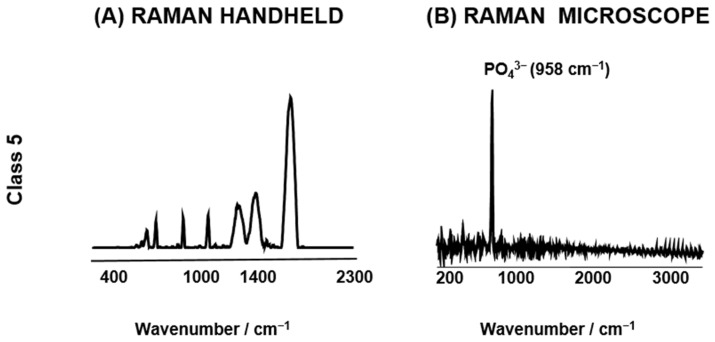
A comparison of the exemplary Raman average spectra of the (**A**) Raman handheld device and the (**B**) Raman microscope of PMI class 5. This particular class yielded less usable results due to the very high fluorescence. The analysis presented in Figure 2 indicates that both the handheld Raman device and the Raman microscope are not suitable for PMI class 5.

**Figure 3 bioengineering-11-01151-f003:**
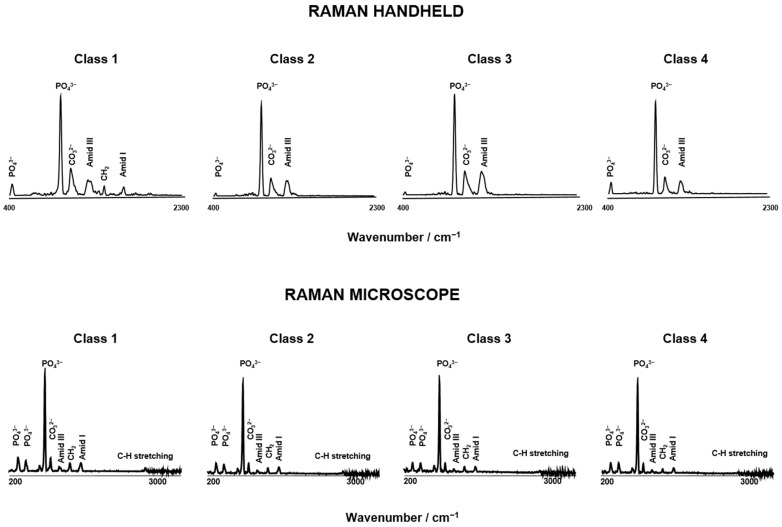
Comparison of mean spectra of human bone samples from Mira Raman handheld device and Senterra Raman microscope of PMI class 1 to 4.

**Figure 4 bioengineering-11-01151-f004:**
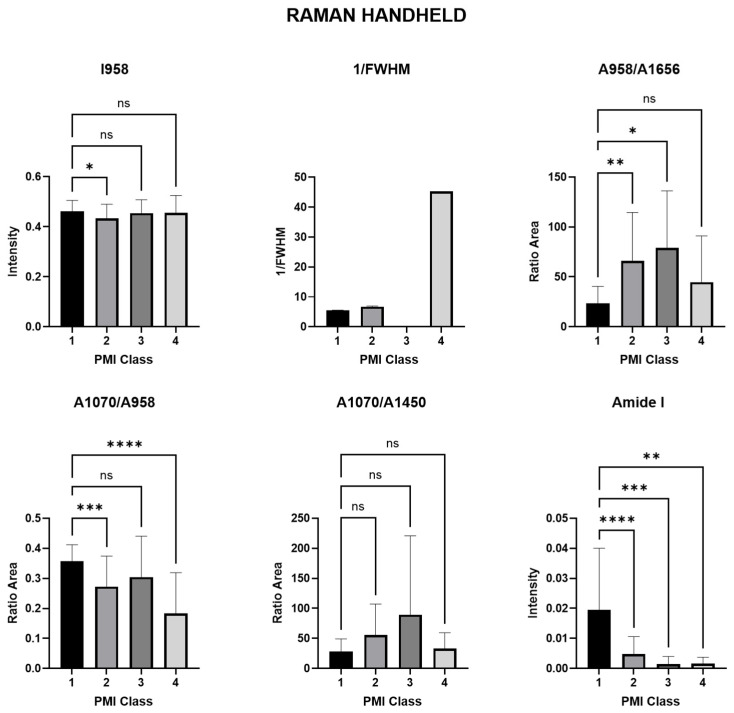
A bar graph depicting spectral markers derived from handheld Raman spectroscopy for the characterization of human bone. The symbol * (*p* < 0.05) indicates statistical significance, ** (*p* < 0.01) represents high significance, *** (*p* < 0.001) signifies very high significance, and **** (*p* < 0.0001) denotes the utmost significance of the data. ns indicates that the difference between the groups was not statistically significant (*p* > 0.05).

**Figure 5 bioengineering-11-01151-f005:**
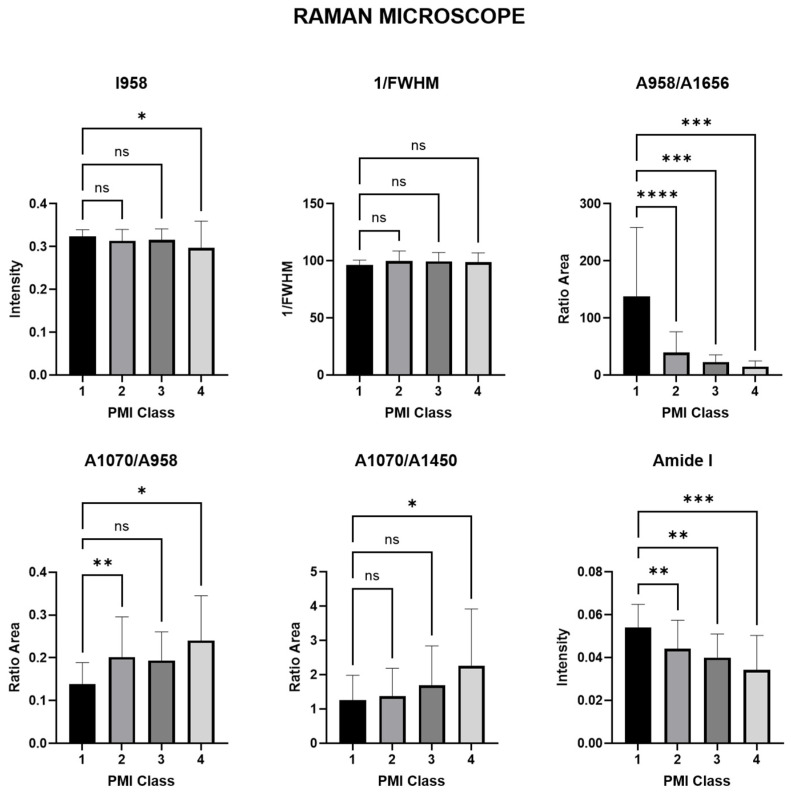
A bar graph depicting spectral markers derived from Raman microscope for the characterization of human bone. The symbol * (*p* < 0.05) indicates statistical significance, ** (*p* < 0.01) represents high significance, *** (*p* < 0.001) signifies very high significance, and **** (*p* < 0.0001) denotes the utmost significance of the data. ns indicates that the difference between the groups was not statistically significant (*p* > 0.05).

**Figure 6 bioengineering-11-01151-f006:**
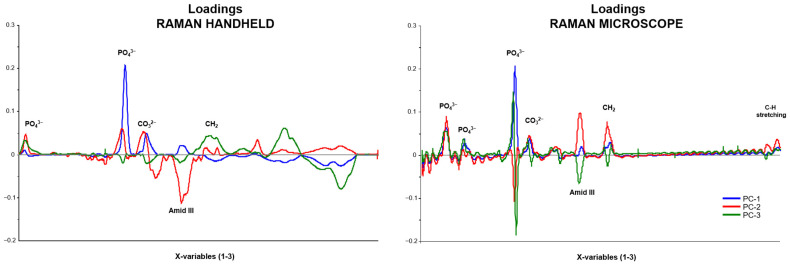
A plot is available to showcase the loadings of Raman handheld and microscope PC-1 (blue), PC-2 (red), and PC-3’s (green) main components.

**Figure 7 bioengineering-11-01151-f007:**
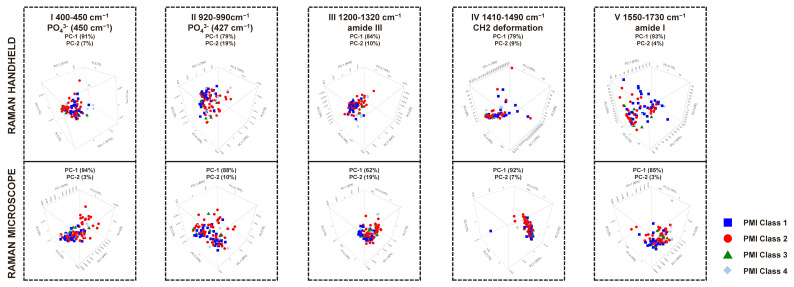
The PCA for the entire measured range (400 cm^−1^ to 2300 cm^−1^ for the Raman handheld device—upper row—and 250 cm^−1^ to 3600 cm^−1^ for the Raman microscope—lower row), with the blue, red, green and light blue symbols representing the respective PMI classes I to IV of the human bone samples.

**Table 1 bioengineering-11-01151-t001:** Spectral markers were obtained from Raman handheld analysis to characterize human bone. These markers, based on band intensity (I), band area (A), and full width at half maximum (FWHM), have relevant applications within medical diagnostics. Their diagnostic importance is determined using a standard one-way ANOVA, where * (*p* < 0.05) indicates statistical significance, ** (*p* < 0.01) reflects high significance, *** (*p* < 0.001) signifies very high significance, and **** (*p* < 0.0001) denotes extremely high significance. ^ns^ indicates that the difference between the groups was not statistically significant (*p* > 0.05).

Name	Description	Determination	*p*-Values Derived from a Single-Factor ANOVA		
			Class 1 vs. 2	Class 1 vs. 3	Class 1 vs. 4
Phosphate	ν_1_PO_4_^3−^ Amount of phosphate	(I_958_)	0.0479 *	0.9553 ^ns^	0.9731 ^ns^
Crystallinity (CI)	mineral quality-crystallinity index	1/FWHM_958_	0.0154 *	0.5954 ^ns^	0.5195 ^ns^
Mineral/matrix (MMR) phosphate/amide I	ν_1_PO_4_^3−^/amide I Mineral component amount to the organic one	(A_958_/A_1656_)	0.0013 **	0.0458 *	0.6382 ^ns^
Mineral quality and crystallinity carbonate/phosphate	ν_1_CO_3_^2−^/ν_1_PO_4_^3−^ Carbonate incorporation extent in the hydroxyapatite lattice	(A_1070_/A_958_)	0.0008 ***	0.3056 ^ns^	<0.0001 ****
Mineral carbonate content (MinCarb)	ν_1_CO_3_^2−^/(C-H) bend; CH_2_ wag	(A_1070_/A_1450_)	0.1292 ^ns^	0.0686 ^ns^	0.9960 ^ns^
amide I	amide I of α-helical structures Arrangement and quantity of collagen	(I_1656_)	<0.0001 ****	0.0008 ***	0.0026 **

**Table 2 bioengineering-11-01151-t002:** Spectral markers were obtained from Raman microscope analysis to characterize human bone. These markers, based on band intensity (I), band area (A), and full width at half maximum (FWHM), have relevant applications within medical diagnostics. Their diagnostic importance is determined using a standard one-way ANOVA, where * (*p* < 0.05) indicates statistical significance, ** (*p* < 0.01) reflects high significance, *** (*p* < 0.001) signifies very high significance, and **** (*p* < 0.0001) denotes extremely high significance. ^ns^ indicates that the difference between the groups was not statistically significant (*p* > 0.05).

Name	Description	Determination	*p*-Values Derived from a Single-Factor ANOVA		
			Class 1 vs. 2	Class 1 vs. 3	Class 1 vs. 4
Phosphate	ν_1_PO_4_^3−^ Amount of phosphate	(I_958_)	0.2948 ^ns^	0.7299 ^ns^	0.0347 *
Crystallinity (CI)	mineral quality-crystallinity index	1/FWHM_958_	0.0977 ^ns^	0.4968 ^ns^	0.7418 ^ns^
Mineral/matrix (MMR) phosphate/amide I	ν_1_PO_4_^3−^/amide I Mineral component amount to the organic one	(A_958_/A_1656_)	<0.0001 ****	0.0003 ***	0.0010 ***
Mineral quality and crystallinity carbonate/phosphate	ν_1_CO_3_^2−^/ν_1_PO_4_^3−^ Carbonate incorporation extent in the hydroxyapatite lattice	(A_1070_/A_958_)	0.0087 **	0.2425 ns	0.0103 *
Mineral carbonate content (MinCarb)	ν_1_CO_3_^2−^/(C-H) bend; CH_2_ wag	(A_1070_/A_1450_)	0.9385 ^ns^	0.4849 ^ns^	0.0202 *
amide I	amide I of α-helical structures Arrangement and quantity of collagen	(I_1656_)	0.0026 **	0.0060 **	0.0002 ***

**Table 3 bioengineering-11-01151-t003:** PCA of 1 to 4 PMI classes with Raman handheld device and Raman microscope were compared. Figure 7 displays the PCA plots.

Wave Number Range Number	Handheld PCA	Microscope PCA	Assignment	Spectral Region
I	PC-1 (91%) PC-2 (7%)	PC-1 (94%) PC-2 (3%)	_ν2_PO_4_^3−^ (450 cm^−1^)	400 cm^−1^ to 450 cm^−1^
II	PC-1 (79%) PC-2 (19%)	PC-1 (88%) PC-2 (10%)	_ν1_PO_4_^3−^ (958 cm^−1^)	920 cm^−1^ to 990 cm^−1^
III	PC-1 (84%) PC-2 (10%)	PC-1 (62%) PC-2 (19%)	amide III (1246 cm^−1^)	1200 cm^−1^ to 1320 cm^−1^
IV	PC-1 (79%) PC-2 (9%)	PC-1 (92%) PC-2 (7%)	CH_2_ Deformation (1450 cm^−1^)	1410 cm^−1^ to 1490 cm^−1^
V	PC-1 (93%) PC-2 (4%)	PC-1 (85%) PC-2 (3%)	amide I (1656 cm^−1^)	1550 cm^−1^ to 1730 cm^−1^

## Data Availability

The data presented in this study are available upon request from the corresponding author.

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
