# Peer review of "Raman Handheld Versus Microscopic Spectroscopy for Estimating the Post-Mortem Interval of Human Bones: A Comparative Pilot Study"

_bioengineering, 2024, doi:10.3390/bioengineering11111151_

Round 1
Reviewer 1 Report
Comments and Suggestions for Authors
Title
The title of the article fully reflects the essence of the work carried out.
Abstract
I suggest to shorten the Abstract. In its present form it is hard to read and to understand the essence of the work.
Introduction
Lines 81-83. “separate taphonomic data from archaeological and palaeontological studies are used to assess longer PMIs”. Please, be specific and specify what does it mean “longer” – hours, years, centuries, a thousand years…
The authors claim that in ref. [38] information about parameters obtained with Raman spectroscopy is gathered. In this connection, the originality and novelty of he work should be spelled out more clearly. There should be a statement of the problem and approaches for its solution. As for me, the text in lines 118-121 is not good enough to catch the essential of the paper. I suggest to extend the Introduction.
Materials and Methods
In general, the materials and methods are well structured and described.
Results
The results and their statistical treatments are well described.
Discussion
This section indeed discusses the difference between the handheld instrument and Raman microscope. It gives an understanding of the work that's been done.
Conclusions
This is a very poorly written section. Why conduct the extensive research described in the paper for the sake of stating the understandable fact that good equipment is better than the handheld one? This Section should be fully re-written.
Author Response
Comments 1: Title The title of the article fully reflects the essence of the work carried out.
Response: Thank you very much for your positive feedback!
Comments 2: Abstract I suggest to shorten the Abstract. In its present form it is hard to read and to understand the essence of the work.
Response: Thank you very much for your constructive feedback. We have revised the abstract to improve clarity and conciseness, focusing on our study's key findings and significance.
Comments 3: Introduction Lines 81-83. “separate taphonomic data from archaeological and palaeontological studies are used to assess longer PMIs”. Please, be specific and specify what does it mean “longer” – hours, years, centuries, a thousand years…
Response: Thank you for the feedback. We specified the timeline.
Comments 4: The authors claim that in ref. [38] information about parameters obtained with Raman spectroscopy is gathered. In this connection, the originality and novelty of he work should be spelled out more clearly. There should be a statement of the problem and approaches for its solution. As for me, the text in lines 118-121 is not good enough to catch the essential of the paper. I suggest to extend the Introduction.
Response: We expanded the Introduction to clearly state the originality of our approach and the problem our study addresses using Raman spectroscopy.
Comments 5: Materials and Methods In general, the materials and methods are well structured and described.
Response: Thank you very much for your positive feedback!
Comments 6: Results The results and their statistical treatments are well described.
Response: Thank you very much for your positive feedback!
Comments 7: Discussion This section indeed discusses the difference between the handheld instrument and Raman microscope. It gives an understanding of the work that's been done.
Response: Thank you very much for your positive feedback!
Comments 8: Conclusions This is a very poorly written section. Why conduct the extensive research described in the paper for the sake of stating the understandable fact that good equipment is better than the handheld one? This Section should be fully re-written.
Response: We appreciate your valuable feedback on the clarity and structure of our conclusions. Based on your suggestions, we revised the section to highlight better the study’s objectives, its contributions to PMI estimation methods, and the novelty of comparing handheld and microscope-based Raman spectrometers. The revised conclusions focus on these instruments' practical applications and limitations for PMI analysis, underscoring the distinct advantages and unique contributions of Raman spectroscopy in forensic contexts.
Reviewer 2 Report
Comments and Suggestions for Authors
This article evaluates the potential of handheld and microscopic Raman spectroscopy devices in estimating the post-mortem interval (PMI) of human bones. The researchers compared the two devices on their capability to classify bone samples into PMI classes based on spectral data analysis and principal component analysis (PCA).
There are several issues that should be considered before accepting the manuscript.
1. The study acknowledged the ineffectiveness of both Raman devices for samples older than 100 years (class 5) due to high fluorescence. What is the reason for this fluorescence, and why is it not critical for other classes?
2. The presence of bone lesions, such as oncology, osteoporosis or some infections, might alter spectral features, potentially leading to inaccurate PMI classifications. Please discuss this.
3. The results of the measurements may vary significantly across different conditions, particularly in non-controlled outdoor environments (soil composition, pH, moisture, temperature, etc.). Future studies might benefit from simulating diverse environmental conditions or testing the device in field-simulated conditions to assess how they affect Raman data. And of course, it is important in the future to conduct a comparative analysis of the results with measurements using other optical methods (for example, FTIR).
4. It is unclear why the authors chose the PCA method to classify the data. Perhaps other machine learning methods could provide greater efficiency.
5. The authors should discuss how Raman spectroscopy can be integrated into a wider range of forensic tools.
Author Response
Comments 1: The study acknowledged the ineffectiveness of both Raman devices for samples older than 100 years (class 5) due to high fluorescence. What is the reason for this fluorescence, and why is it not critical for other classes?
Response: Thank you for pointing out the fluorescence challenges with PMI class 5 samples. As discussed in the revised text, fluorescence arises from organic degradation products like collagen, which obscure the Raman signal in older samples. This interference is less significant in younger bones due to fewer fluorescent compounds. In future work, we plan to address this by applying fluorescence suppression techniques to improve analysis of older samples. We appreciate your suggestion, which will help broaden the method's applicability.
Comments 2: The presence of bone lesions, such as oncology, osteoporosis or some infections, might alter spectral features, potentially leading to inaccurate PMI classifications. Please discuss this.
Response: We appreciate your observation regarding the potential impact of pathological bone conditions on spectral accuracy. Bone lesions from diseases like osteoporosis, cancer, or infections alter mineral and organic components, affecting key spectral indicators such as MMR, amide bands, and crystallinity. We have expanded the discussion to acknowledge this limitation and suggest that future research should include samples with known pathological conditions. This would allow us to assess how much such variations may affect PMI classification accuracy and develop protocols to account for these factors.
Comments 3: The results of the measurements may vary significantly across different conditions, particularly in non-controlled outdoor environments (soil composition, pH, moisture, temperature, etc.). Future studies might benefit from simulating diverse environmental conditions or testing the device in field-simulated conditions to assess how they affect Raman data. And of course, it is important in the future to conduct a comparative analysis of the results with measurements using other optical methods (for example, FTIR).
Response: Thank you for your suggestion on environmental variability. We agree that external conditions, such as soil composition, pH, moisture, and temperature, can significantly impact RS measurements. We have added a discussion on this limitation and suggested that future studies should simulate these diverse environmental conditions to better understand their effects on Raman data. Additionally, we plan to incorporate comparative analyses with other optical methods like FTIR to validate and potentially enhance the reliability of RS measurements across different environments.
Comments 4: It is unclear why the authors chose the PCA method to classify the data. Perhaps other machine learning methods could provide greater efficiency.
Response: Thank you for your insightful comment. We chose principal component analysis (PCA) for its effectiveness in dimensionality reduction and its suitability for identifying patterns in complex spectral data, which is critical in distinguishing PMI classes based on subtle spectral differences. PCA allows us to capture the most significant variance within the dataset, facilitating a clear understanding of spectral trends across PMI groups.
However, we acknowledge that other machine learning methods, such as support vector machines (SVM) or random forest classifiers, could potentially enhance classification efficiency by leveraging supervised learning. In future work, we plan to explore these and other algorithms further to assess classification accuracy and efficiency in PMI estimation, potentially increasing the robustness of our model. Thank you for this valuable suggestion, which we believe could strengthen the study's methodology in subsequent analyses.
Comments 5: The authors should discuss how Raman spectroscopy can be integrated into a wider range of forensic tools.
Response: Thank you for your suggestion to expand on the applicability of Raman spectroscopy (RS) within broader forensic toolsets. We have revised the discussion section to include how RS can be effectively integrated with complementary forensic methods such as DNA analysis, toxicology, and isotopic studies. This integration highlights the unique contributions of RS in enhancing PMI estimation accuracy and preserving sample integrity. We believe this addition clarifies the practical potential of RS in multidisciplinary forensic investigations and provides a broader context for its utility in various forensic scenarios.
Reviewer 3 Report
Comments and Suggestions for Authors
Dear authors, I found your paper very interesting.
please find just a few my comments below.
-at section 2.3. Raman microscopic spectroscopy
It is better to change it to "micro-Raman spectrtoscopy"
-at line 171
Excel. I think you should also add "microsoft" if it is the case.
Author Response
Comments 1: -at section 2.3. Raman microscopic spectroscopy
It is better to change it to "micro-Raman spectrtoscopy"
Response: We have made the change as suggested. Thank you for the helpful feedback!
Comments 2: -at line 171
Excel. I think you should also add "microsoft" if it is the case.
Response: We have made the change as suggested. Thank you for the helpful feedback!
Thanks again for the valuable comments! Kind regards,
Claudia Wöss, corresponding author
Round 2
Reviewer 1 Report
Comments and Suggestions for Authors
Authors have corrected the text according to my comments to mu satisfaction.